# SHOW-O: ONE SINGLE TRANSFORMER TO UNIFY MULTIMODAL UNDERSTANDING AND GENERATION

**Jinheng Xie**[1†] **Weijia Mao**[1†] **Zechen Bai**[1†] **David Junhao Zhang**[1†] **Weihao Wang**[2]
**Kevin Qinghong Lin**[1] **Yuchao Gu**[1] **Zhijie Chen**[2] **Zhenheng Yang**[2] **Mike Zheng Shou**[1*]

[1] Show Lab, National University of Singapore    [2] ByteDance

## ABSTRACT

We present a unified transformer, *i.e.,* Show-o, that unifies multimodal understanding and generation. Unlike fully autoregressive models, Show-o unifies autoregressive and (discrete) diffusion modeling to adaptively handle inputs and outputs of various and mixed modalities. The unified model flexibly supports a wide range of vision-language tasks including visual question-answering, text-to-image generation, text-guided inpainting/extrapolation, and mixed-modality generation. Across various benchmarks, it demonstrates comparable or superior performance to existing individual models with an equivalent or larger number of parameters tailored for understanding or generation. This significantly highlights its potential as a next-generation foundation model. Code and models are released at https://github.com/showlab/Show-o.

## 1 INTRODUCTION

*"Alone we can do so little; together we can do so much."* – Helen Keller

Over the past few years, significant advancements have blossomed in the two key pillars of multimodal intelligence: understanding and generation (Fig. 1(a) and (b)). For multimodal understanding, Multimodal Large Language Models (MLLMs) like LLaVA (Liu et al., 2024c) have demonstrated exceptional capabilities in vision-language tasks such as visual question-answering (VQA). For the other pillar of visual generation, denoising diffusion probabilistic models (DDPMs) (Sohl-Dickstein et al., 2015; Ho et al., 2020b) have revolutionized the traditional generative paradigms (Kingma & Welling, 2013; Goodfellow et al., 2014), achieving unprecedented performance in text-to-image/video generation (Podell et al., 2023; Esser et al., 2024; Ho et al., 2022; Wu et al., 2023a).

Given these achievements in individual fields, it is natural to explore the potential of connecting them. Recent works (Wu et al., 2023b; Ge et al., 2024; Ye et al., 2024a; Dong et al., 2024) have tried to assemble expert models from different domains to form a unified system that can handle both multimodal understanding and generation. However, existing attempts mainly treat each domain independently and often involve individual models responsible for understanding and generation separately (as shown on the left of Fig. 1(c)). For instance, NExT-GPT (Wu et al., 2023b) employs a base language model for multimodal understanding but requires an additional pre-trained diffusion model for image generation. Nonetheless, the mainstream understanding models like LLaVA are of transformer architecture (Vaswani et al., 2017b) while each leading generation models like Stable Diffusion 3 (SD3) (Esser et al., 2024) are just another transformer. This motivates a research question: **can one single transformer handle both multimodal understanding and generation?**

Very recently, Chameleon (Team, 2024) has demonstrated this is possible. Specifically, Chameleon enables an early fusion of different modalities to generate both text and image tokens through the same manner of autoregressive modeling. While it is reasonable to model text tokens autoregressively (Touvron et al., 2023; Liu et al., 2024c), it is less clear whether it is better to model image/video patches (or pixels) autoregressively as well. An apparent and significant bottleneck of autoregressively predicting an image is the large number of sampling steps required due to its causal attention, particularly when dealing with images/videos in higher resolution. Further, (continuous)

---

[†] Equal Contribution    [*] Corresponding Author

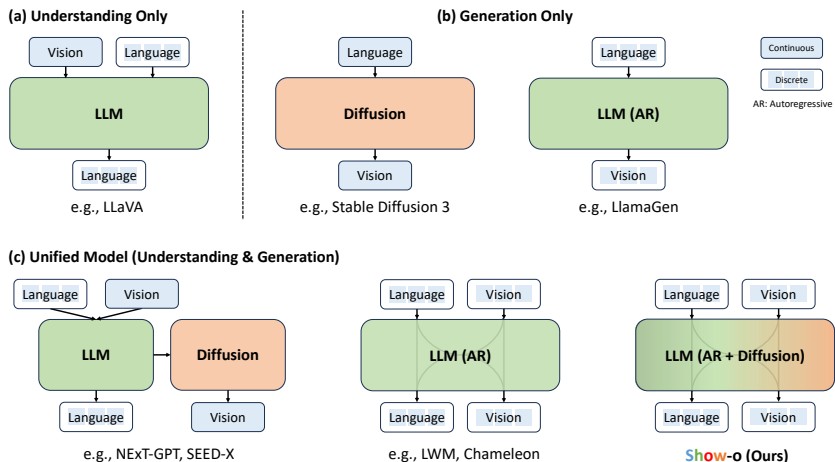

Figure 1: Characteristics comparison among understanding only, generation only, and unified (understanding & generation) models. "Vision" and "Language" indicate the representations from specific input modalities. In this context, "Diffusion" represents both continuous and discrete diffusion.

diffusion models (Podell et al., 2023; Esser et al., 2024) have exhibited superior capabilities in visual generation than autoregressive ones and are in full attention.

This motivates us to ponder: **can such one single transformer involve both autoregressive and diffusion modeling?** Here we envision a new paradigm that text is represented as discrete tokens and modeled autoregressively, same with large language models (LLMs), and continuous image pixels are modeled using denoising diffusion. However, it is non-trivial to integrate these two distinct techniques into one single network due to the significant differences between discrete text tokens and continuous image/video representations. Another challenge lies in the fact that existing state-of-the-art diffusion models typically rely on two distinct models, *i.e.,* a text encoder to encode text conditional information and a denoising network to predict noise.

To this end, we present a novel unified model, *i.e.,* Show-o, capable of addressing both multimodal understanding and generation tasks simultaneously with mixed autoregressive and diffusion modeling (as shown in Fig. 2). Specifically, Show-o is built upon a pre-trained LLM and inherits the autoregressive modeling capability for text-based reasoning. Inspired by Gu et al. (2022); Chang et al. (2022), we employ a simplified discrete denoising diffusion, similar to MaskGIT (Chang et al., 2022), to model discrete image tokens instead of continuous diffusion used in existing works (Ge et al., 2024; Dong et al., 2024). Besides, Show-o inherently encodes text conditional information, eliminating additional text encoders. To accommodate diverse input data and variations of tasks, a text tokenizer and image tokenizer are employed to encode them into discrete tokens, and a unified prompting strategy is proposed further to process these tokens into structure sequences as input. Consequently, given an image accompanying questions, Show-o gives the answers autoregressively. When provided only text tokens, Show-o generates images in a style of discrete denoising diffusion.

Quantitatively, Show-o demonstrates comparable even better performance to individual models with an equivalent or larger number of parameters across benchmarks. In contrast to autoregressively generating an image, Show-o requires approximately 20 times fewer sampling steps, exhibiting inherent potential in acceleration. Besides, as shown in Fig. 2, Show-o naturally supports various downstream applications like text-guided inpainting and extrapolation, without any fine-tuning. Moreover, we have demonstrated that Show-o has the potential for mixed-modality generation like interleaved video keyframe generation with text descriptions, video understanding, and video generation. This demonstrates the potential of the unified model as a feasible paradigm for long-form video understanding and generation. Beyond, we investigate the impact of dataset scale, image resolution, and different types of image representations (discrete or continuous) on the multimodal understanding performance, presenting systematic insights for the design of a unified model in the future.

In Fig. 1, we present a comparison of model characteristics between Show-o and existing representative methods across various domains. One can observe that Show-o is a unified model that flexibly involves existing advanced techniques to comprehensively address multimodal understanding and generation. Collectively, the main contributions of this paper can be summarized as:

- We present a unified model, *i.e.,* Show-o, which unifies multimodal understanding and generation using one single transformer.

- Show-o innovatively unifies autoregressive and (discrete) diffusion modeling within one single transformer, demonstrating versatility in handling both text and images distinctly.

- As a unified model, Show-o demonstrates comparable even better performance to individual baseline models with an equivalent or larger number of parameters in multimodal understanding and generation benchmarks.

- Show-o inherently supports various downstream applications like text-based inpainting and extrapolation, without necessitating any fine-tuning. Besides, it also demonstrates the potential for mixed-modality generation, video understanding, and video generation.

- We explore the impact of dataset scale, image resolution, and different types of representations (discrete or continuous) on multimodal understanding, providing valuable insights for improving multimodal understanding capabilities of a unified model.

## 2 RELATED WORK

### 2.1 MULTIMODAL UNDERSTANDING

Significant advancements in large language models (LLMs) (Touvron et al., 2023; Brown et al., 2020; Chowdhery et al., 2023) have inspired the development of multimodal large language models (MLLMs) (Li et al., 2024; Yin et al., 2023; Bai et al., 2024). Early MLLM efforts, such as LLaVA (Liu et al., 2024c), MiniGPT-4 (Zhu et al., 2023a), and InstructBLIP (Dai et al., 2023), demonstrate notable multimodal understanding capabilities. To integrate LLMs into multimodal domains, these studies explored projecting features from a pre-trained modal-specific encoder, such as CLIP (Radford et al., 2021), into the input space of LLMs, enabling multimodal understanding and reasoning within the transformer backbone. There are various design choices of MLLM (McKinzie et al., 2024; Tong et al., 2024) in vision encoders, feature alignment adapters, and datasets.

### 2.2 VISUAL GENERATION

**Autoregressive models.** Transformer models (Vaswani et al., 2017a; Raffel et al., 2020; Brown et al., 2020; Touvron et al., 2023) have demonstrated great success of autoregressive modeling in natural language processing. Inspired by such progress, previous studies (Parmar et al., 2018; Esser et al., 2021; Ravuri & Vinyals, 2019; Chen et al., 2020; Kondratyuk et al., 2023) directly apply the same autoregressive modeling to learn the dependency of image pixels for image/video generation. For instance, VideoPoet (Kondratyuk et al., 2023) also employs the decoder-only transformer architecture for synthesizing high-quality videos from multimodal inputs. More recently, LlamaGen (Sun et al., 2024) has demonstrated LLM-architecture based image token autoregression.

**Diffusion models.** In recent years, diffusion-based methods (Rombach et al., 2022; Ramesh et al., 2022b;a; Peebles & Xie, 2023; Bao et al., 2023; Podell et al., 2023; Chen et al., 2024; Nichol et al., 2021; Xue et al., 2024; Xie et al., 2023; Wu et al., 2023a) have demonstrated exceptional capabilities in text-to-image/video generation. Typically, the denoising diffusion process is operated on the continuous latent space, in which the model is tasked with predicting the added Gaussian noise. In contrast, D3PM (Austin et al., 2021), Mask-predict (Ghazvininejad et al., 2019), ARDM (Hoogeboom et al., 2022), MaskGIT (Chang et al., 2022), UniD3 (Hu et al., 2023), and Copilot4D (Zhang et al., 2024) formulate a discrete corruption process as an alternative to Gaussian diffusion.

### 2.3 UNIFIED VISION-LANGUAGE FOUNDATION MODEL

In recent years, an increasing number of studies (Wu et al., 2023b; Tang et al., 2024; Ye et al., 2024a; Aiello et al., 2024; Lu et al., 2024) have focused on unified multimodal language models capable of both comprehension and generation. Some efforts (Zhu et al., 2023b; Sun et al., 2023b;a) use continuous representations interleaved with text tokens for autoregressive modeling to generate images. SEED-X (Ge et al., 2024) proposes a unified and versatile foundation system capable of handling both multimodal understanding and generation tasks. DreamLLM (Dong et al., 2024) also explores

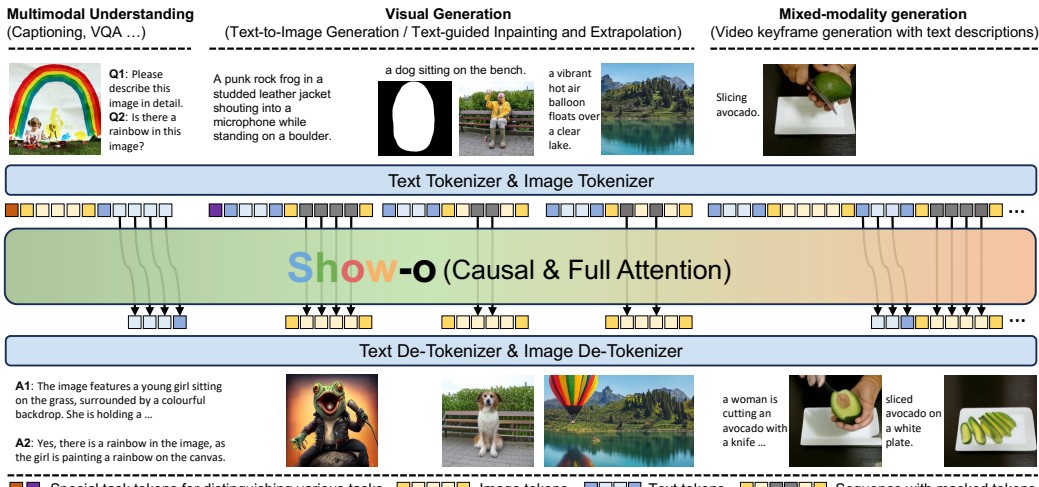

Figure 2: An overview of Show-o. The input data, regardless of its modalities, is tokenized and then prompted into a formatted input sequence. Show-o processes text tokens autoregressively with causal attention and image tokens in (discrete) diffusion modeling via full attention, and then generates the desired output. Specifically, Show-o can handle image captioning, visual question answering, text-to-image generation, text-guided inpainting/extrapolation, and mixed modality generation.

the potential of enabling multimodal comprehension and creation. Chameleon (Team, 2024) introduces token-based mixed-modal models capable of comprehending and generating images.

# 3 METHODOLOGY

**Preliminaries.** Instead of continuous diffusion, this work employs mask token prediction used in MaskGIT as a simplified discrete diffusion modeling to enable a more unified learning objective, *i.e.*, predicting discrete tokens within one single transformer. We draw the connection between mask token prediction used in this work and discrete diffusion modeling in Appendix A.

## 3.1 TOKENIZATION

Show-o is built upon pre-trained LLMs (Li et al., 2023), it is natural to perform the unified learning on the discrete space. We maintain a unified vocabulary to include discrete text and image tokens.

**Text Tokenization.** Show-o is based on a pre-trained LLM such that we utilize the same tokenizer for text data tokenization without any modifications.

**Image Tokenization.** Following MAGVIT-v2 (Yu et al., 2023), we train a lookup-free quantizer using a large-scale image data. The quantizer maintains a codebook of size $K = 8,192$ and encodes images of $256\times256$ resolution into $16\times16$ discrete tokens (option (a) in Fig. 3).

An alternative approach is to use different tokenizers for understanding and generation, respectively. Inspired by existing studies (Liu et al., 2024c;b), we also extract the continuous image representations from the pre-trained MAGVIT-v2 and CLIP-ViT (Radford et al., 2021) encoder as input for exploring the improvement of multimodal understanding capabilities (options (b) and (c) in Fig. 3). We will present more details and discuss this exploration in Section 4.6. In the following sections, the default Show-o employs discrete image tokens as input for both multimodal understanding and generation (option (a) in Fig. 3). For simplicity, we only elaborate on the default Show-o in the methodology sections.

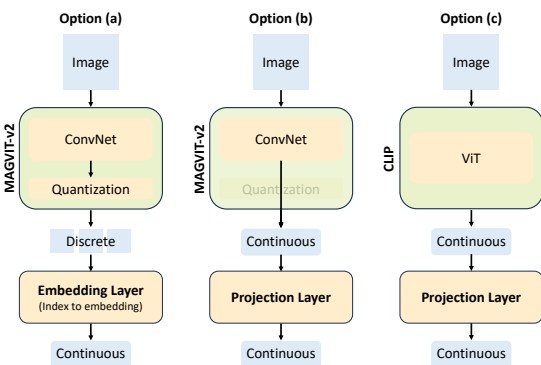

Figure 3: Optional inputs for multimodal understanding.

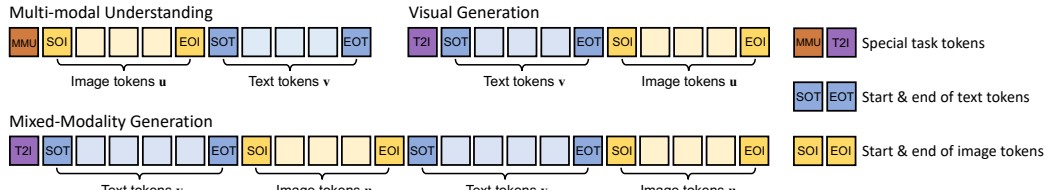

Figure 4: Illustration of the proposed unified prompting format.

## 3.2 ARCHITECTURE

Show-o inherits the architecture of existing LLM (Li et al., 2023) without any architecture modifications except for prepending a QK-Norm operation (Dehghani et al., 2023; Wortsman et al., 2023; Team, 2024) to each attention layer. We initialize Show-o with the weights of a pre-trained LLM and expand the size of the embedding layer by incorporating 8,192 new learnable embeddings for discrete image tokens. Unlike state-of-the-art diffusion models that require an additional text encoder, Show-o inherently encodes text conditional information by itself for text-to-image generation.

**Unified Prompting.** To perform unified learning on multimodal understanding and generation, we design a unified prompting strategy to format various kinds of input data. Given an image-text pair $(\mathbf{x}, \mathbf{y})$, it is first tokenized into $M$ image tokens $\mathbf{u} = \{u_i\}_{i=1}^{M}$ and $N$ text tokens $\mathbf{v} = \{v_i\}_{i=1}^{N}$ by the image and text tokenizer, respectively. We form them into an input sequence according to the type of task in the format illustrated in Fig. 4. Specifically, [MMU] and [T2I] are pre-defined task tokens that indicate the learning task for the input sequence. [SOT] and [EOT] serve as special tokens denoting the start and end of text tokens, respectively. Similarly, [SOI] and [EOI] are pre-defined special tokens marking the start and end of image tokens.

By employing this prompt design, we can effectively encode various input data for multi-modal understanding, text-to-image generation, and mixed-modality generation as sequential data. This setup enables unified learning to operate seamlessly within sequences across these various tasks. Once trained, we can accordingly prompt Show-o to handle various vision-language tasks including visual question answering and text-to-image generation (as shown in Fig. 2).

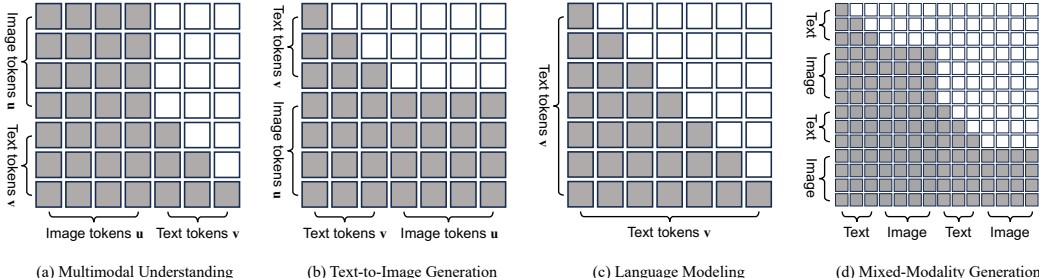

(a) Multimodal Understanding   (b) Text-to-Image Generation   (c) Language Modeling   (d) Mixed-Modality Generation

Figure 5: Omni-Attention Mechanism (The dark squares represent 'allow to attend', while the white squares indicate 'prevent from attending'). It is a versatile attention mechanism with causal and full attention that adaptively mixes and changes according to the format of the input sequence.

**Omni-Attention Mechanism.** Different from existing works (Touvron et al., 2023; Team, 2024) that model sequence auto-regressively only, we propose an omni-attention mechanism to enable Show-o to model various types of signals in distinct ways. It is a comprehensive attention mechanism with causal and full attention that adaptively mixes and changes according to the format of the input sequence. We illustrate omni-attention examples for different input sequences in Fig. 5. Specifically, Show-o model text tokens $\mathbf{v}$ within the sequence via causal attention. For image tokens $\mathbf{u}$, Show-o processes them via full attention, allowing each token to comprehensively interact with all others. Given a formatted input sequence, it is apparent that in multimodal understanding (Fig. 5(a)), text tokens in a sequence can attend to all previous image tokens, and in text-to-image generation (Fig. 5(b)), image tokens are able to interact with all preceding text tokens. When given only text tokens, it degrades to causal attention (Fig. 5(c)).

**Training Objectives.** To perform both auto-regressive and (discrete) diffusion modeling, we employ two learning objectives: i) Next Token Prediction (NTP) and ii) Mask Token Prediction (MTP). Given a sequence with $M$ image tokens $\mathbf{u} = \{u_1, u_2, \cdots, u_M\}$ and $N$ text tokens

Table 1: Evaluation on multimodal understanding benchmarks. Show-o is currently built upon Phi-1.5 and thus we implement LLaVA-v1.5-Phi-1.5 as our apple-to-apple baseline. Und. and Gen. denote "understanding" and "generation", respectively. ‡ denotes the improved Show-o that employs CLIP-ViT continuous representations. We highlight the model size of Show-o and LLaVA baseline in green, and we use blue to highlight the larger model size than ours.

| Type | Model | # Params | POPE↑ | MME↑ | Flickr30k↑ | VQAv2$_{\text{(test)}}$↑ | GQA↑ | MMMU↑ |
|------|-------|----------|-------|------|-----------|--------------------------|------|-------|
| Und. Only | LLaVA-v1.5 (Liu et al., 2024b) | 7B | 85.9 | 1510.7 | - | 78.5 | 62.0 | 35.4 |
| | InstructBLIP (Dai et al., 2023) | 13B | 78.9 | 1212.8 | - | - | 49.5 | - |
| | Qwen-VL-Chat Bai et al. (2023) | 7B | - | 1487.5 | - | 78.2 | 57.5 | - |
| | mPLUG-Owl2 (Ye et al., 2024b) | 7B | 85.8 | 1450.2 | - | 79.4 | 56.1 | - |
| | LLaVA-v1.5-Phi-1.5 | 1.3B | 84.1 | 1128.0 | 69.6 | 75.3 | 56.5 | 30.7 |
| Und. and Gen. | Gemini-Nano-1 (Anil et al., 2023) | 1.8B | - | - | - | 62.7 | - | 26.3 |
| | CoDI (Tang et al., 2024) | - | - | - | 12.8 | - | - | - |
| | Emu (Sun et al., 2023c) | 13B | - | - | 77.4 | 57.2 | - | - |
| | NExT-GPT (Wu et al., 2023b) | 13B | - | - | 84.5 | 66.7 | - | - |
| | SEED-X (Ge et al., 2024) | 17B | 84.2 | 1435.7 | 52.3 | - | 47.9 | 35.6 |
| | DreamLLM (Dong et al., 2024) | 7B | - | - | - | 72.9 | - | - |
| | VILA-U (Wu et al., 2024) | 7B | 85.8 | 1401.8 | - | 79.4 | 60.8 | 31.6 |
| | Emu3 (Wang et al., 2024) | 8B | 85.2 | - | - | 75.1 | 60.3 | - |
| | Chameleon (Team, 2024) | 34B | - | - | 74.7 | 66.0 | - | - |
| | **Show-o (Ours)** | 1.3B | 80.0 | 1097.2 | 62.5 | 69.4 | 58.0 | 26.7 |
| | **Show-o$^{\ddagger}$ (Ours)** | 1.3B | 84.5 | 1232.9 | 67.6 | 74.7 | 61.0 | 27.4 |

$\mathbf{v} = \{v_1, v_2, \cdots, v_N\}$ for multimodal understanding, we maximize the likelihood of text tokens by employing the standard language modeling objective:

$$\mathcal{L}_{\text{NTP}} = \sum_i \log p_\theta(v_i | v_1, \cdots, v_{i-1}, u_1, \cdots, u_M), \tag{1}$$

where $p(\cdot|\cdot)$ indicates the conditional probability which is modeled by the weights $\theta$ of Show-o and stochastic gradient descent is used to train the model. Note that, if the input sequence involves only text tokens, there are no conditional terms on image tokens $\mathbf{u} = \{u_1, u_2, \cdots, u_M\}$.

With the proof in Appendix A, we seamlessly integrate the simplified discrete diffusion modeling within Show-o by employing the mask token prediction as a learning objective. Hence, for modeling image tokens $\mathbf{u} = \{u_1, u_2, \cdots, u_M\}$ within the input sequence, we first randomly replace the image tokens with the [MASK] token, notated as $u_*$, at a random ratio (controlling by a time step) to create a masked sequence $\mathbf{u}_* = \{u_*, u_2, \cdots, u_*, u_M\}$. An illustration can be found in Fig. 9. Next, we aim to reconstruct the original image token from the masked tokens conditioning on unmasked regions and preceding text tokens by maximizing the following likelihood:

$$\mathcal{L}_{\text{MTP}} = \sum_j \log p_\theta(u_j | u_*, u_2, \cdots, u_*, u_M, v_1, \cdots, v_N). \tag{2}$$

Note that the loss is only applied to the masked tokens. Specifically, we follow the sampling strategy used by MaskGIT Chang et al. (2022; 2023) to mask image tokens and reconstruct them via the information from all text and unmasked image tokens within the input sequence. Following the classifier-free guidance introduced by Ho & Salimans (2022), we randomly replace the conditioned text tokens using a null text "" with some probability.

Given a batch size of input sequences, the overall training loss is the combination of $\mathcal{L}_{\text{MTP}}$ and $\mathcal{L}_{\text{NTP}}$:

$$\mathcal{L} = \mathcal{L}_{\text{MTP}} + \alpha \mathcal{L}_{\text{NTP}}, \tag{3}$$

where $\alpha$ is the hyper-parameter weighting the loss term $\mathcal{L}_{\text{NTP}}$. The training schedule mainly involves three stages, and we provide more details in Appendix C.

**Inference Stage.** In multimodal understanding, given an image accompanying visual questions, Show-o autoregressively predicts textual answers. In visual generation, we use all [MASK] tokens as initial input for Show-o, in which [MASK] tokens will be iteratively replaced by the predicted image tokens within $T$ steps. More inference details are provided in Appendix D.

## 4 EXPERIMENTS

### 4.1 EXPERIMENTAL SETUP

**Datasets.** We assemble two scales of datasets, *i.e.,* around 35M and 2.0B image-text pairs, and collect around 2M high-quality data for multimodal understanding and generation fine-tuning. Besides,

**User**: Do you think the image is unusual or not?

**Show-o**:
The image is unusual because it depicts a living room with a couch and a table in the middle of the ocean, surrounded by a large body of water. This is not a typical setting for a living room, as living rooms are usually indoors and designed for relaxation and entertainment. The presence of the couch and table in the ocean adds a unique and unexpected element to the scene, making it stand out from typical living room setups.

**Chameleon**:
The image is unusual because of the transparent and colorful fish and coral in the water. The image is also unusual because of the clear blue sky with white clouds. The unusual objects in the image are the fish and coral in the water. The unusual objects in the image are the fish and coral in the water. The unusual objects in the image are the fish and coral in the water. The unusual objects in the image are the fish and coral in the water.

Figure 6: Comparisons of VQA capabilities between Chameleon and Show-o.

RefinedWeb (Penedo et al., 2023) is adopted as text corpora to maintain the language modeling capability. Appendix E provides more details about these datasets.

**Evaluation Details.** Following LLaVA (Liu et al., 2024b), we evaluate the multimodal understanding capabilities of Show-o on POPE, MME, Flickr30k, VQAv2, GQA, and MMMU benchmarks. Besides, we adopt Fréchet Inception Distance (FID) on MSCOCO dataset to evaluate the generation fidelity of Show-o. Further, we follow SD3 (Esser et al., 2024) to evaluate the text-to-image generation capabilities of Show-o on the GenEval (Ghosh et al., 2023) benchmark.

**Implementation details.** Current version of Show-o is based on Phi-1.5 (1.3B) (Li et al., 2023). In the following, the default Show-o employs discrete image tokens as input for both multimodal understanding and generation. Show-o$^\dagger$ and Show-o$^\ddagger$ indicate the use of continuous image representations from the pre-trained MAGVIT-v2 and CLIP-ViT (corresponding to options (b) and (c) in Fig. 3), respectively, for multimodal understanding. Training details can be found in Appendix F.

## 4.2 MULTIMODAL UNDERSTANDING

**Quantitative Evaluation.** Table 1 presents the multimodal understanding capability of Show-o on public benchmarks, such as image captioning and visual question-answering tasks. i) The current version of Show-o is built upon Phi-1.5 and thus we follow LLaVA to train Show-o's understanding only counterpart as our direct baseline, namely LLaVA-v1.5-Phi-1.5. The proposed Show-o exhibits comparable performance in all evaluation metrics to the baseline LLaVA-v1.5-Phi-1.5, which is dedicated and optimized to only multimodal understanding. This demonstrates the great potential of our framework to unify multimodal understanding and generation in one single transformer. ii) When comparing with understanding only models including InstructBLIP, Qwen-VL-Chat, and mPLUG-Owl2 on multimodal understanding, our model with a much smaller model size also achieves competitive performance on POPE, MME, Flickr30k and VQAv2 benchmarks and performs better on GQA benchmark. iii) Compared with unified models with a much larger number of parameters, such as NExT-GPT-13B and Chameleon-34B, our model also achieves decent performance on Flickr30k benchmark and performs much better on VQAv2 benchmark.

**Qualitative Results.** We present Show-o's visual question-answering capability and make comparisons with Chameleon in Fig. 6. It is evident that when presented with a query image, Show-o can respond to commonly asked questions, even addressing the unusual aspects within the image. In the example of Fig. 6, when asked, "Do you think the image is unusual or not", Chameleon fails to correctly identify the unusual aspect. In contrast, Show-o's response, "as living rooms are **usually indoors** and designed for relaxation and entertainment", is more accurate.

## 4.3 VISUAL GENERATION

**Results on MSCOCO 30K.** We present zero-shot FID of Show-o on MSCOCO 30K in Table 2. It can be observed that, compared to generation models trained with larger numbers of parameters and training images such as GLIDE and DALL·E 2, Show-o achieves a better FID, *i.e.,* 9.24, with only 1.3B parameters and 35M training data. Though Giga-GAN, Imagen, and RAPHAEL obtain a relatively better performance

Table 2: MSCOCO zero-shot FID. Und. and Gen. denote "understanding" and "generation", respectively.

| Type | Method | # Params | # Images | FID-30K↓ |
|------|--------|----------|----------|----------|
| Gen. Only | DALL·E (Ramesh et al., 2021) | 12B | 250M | 27.50 |
| | GLIDE (Nichol et al., 2021) | 5B | 250M | 12.24 |
| | LDM (Rombach et al., 2022) | 1.4B | 400M | 12.64 |
| | DALL·E 2 (Ramesh et al., 2022a) | 6.5B | 650M | 10.39 |
| | SDv1.5 (Rombach et al., 2022) | 0.9B | 2000M | 9.62 |
| | GigaGAN (Kang et al., 2023) | 0.9B | 2700M | 9.09 |
| | PixArt (Chen et al., 2024) | 0.6B | 25M | 7.32 |
| | Imagen (Saharia et al., 2022) | 3B | 860M | 7.27 |
| | RAPHAEL (Xue et al., 2024) | 3B | 5000M+ | 6.61 |
| Und. and Gen. | CoDI (Tang et al., 2024) | - | 400M | 22.26 |
| | LWM (Liu et al., 2024a) | 7B | - | 12.68 |
| | SEED-X (Ge et al., 2024) | 17B | - | 14.99 |
| | DreamLLM (Dong et al., 2024) | 7B | - | 8.76 |
| | **Show-o (Ours)** | 1.3B | 35M | 9.24 |

Table 3: Evaluation on the GenEval (Ghosh et al., 2023) benchmark. Und. and Gen. denote "understanding" and "generation", respectively. We highlight the model size of Show-o in green, and we use blue to highlight the larger model size than ours. Obj.: Object. Attri.: Attribute.

| Type | Method | # Params | Single Obj. | Two Obj. | Counting | Colors | Position | Color Attri. | Overall↑ |
|---|---|---|---|---|---|---|---|---|---|
| Gen. Only | LlamaGen (Sun et al., 2024) | 0.8B | 0.71 | 0.34 | 0.21 | 0.58 | 0.07 | 0.04 | 0.32 |
| | LDM (Rombach et al., 2022) | 1.4B | 0.92 | 0.29 | 0.23 | 0.70 | 0.02 | 0.05 | 0.37 |
| | SDv1.5 (Rombach et al., 2022) | 0.9B | 0.97 | 0.38 | 0.35 | 0.76 | 0.04 | 0.06 | 0.43 |
| | PixArt-alpha (Chen et al., 2024) | 0.6B | 0.98 | 0.50 | 0.44 | 0.80 | 0.08 | 0.07 | 0.48 |
| | SDv2.1 (Rombach et al., 2022) | 0.9B | 0.98 | 0.51 | 0.44 | 0.85 | 0.07 | 0.17 | 0.50 |
| | DALL-E 2 (Ramesh et al., 2022a) | 6.5B | 0.94 | 0.66 | 0.49 | 0.77 | 0.10 | 0.19 | 0.52 |
| | SDXL (Podell et al., 2023) | 2.6B | 0.98 | 0.74 | 0.39 | 0.85 | 0.15 | 0.23 | 0.55 |
| | SD3 (d=24) (Esser et al., 2024) | 2B | 0.98 | 0.74 | 0.63 | 0.67 | 0.34 | 0.36 | 0.62 |
| Und. and Gen. | CoDI (Tang et al., 2024) | - | 0.89 | 0.16 | 0.16 | 0.65 | 0.02 | 0.01 | 0.31 |
| | LWM (Liu et al., 2024a) | 7B | 0.93 | 0.41 | 0.46 | 0.79 | 0.09 | 0.15 | 0.47 |
| | SEED-X (Ge et al., 2024) | 17B | 0.97 | 0.58 | 0.26 | 0.80 | 0.19 | 0.14 | 0.49 |
| | Emu3 Wang et al. (2024) | 8B | - | - | - | - | - | - | 0.66 |
| | Transfusion Zhou et al. (2024) | 7.3B | - | - | - | - | - | - | 0.63 |
| | Chameleon (Team, 2024) | 7B | - | - | - | - | - | - | 0.39 |
| | **Show-o (Ours)** | 1.3B | 0.98 | 0.80 | 0.66 | 0.84 | 0.31 | 0.50 | 0.68 |
| | **Show-o‡ (Ours)** | 1.3B | 0.98 | 0.85 | 0.67 | 0.81 | 0.28 | 0.55 | 0.69 |

than Show-o, they are much larger in model size (3B v.s. 1.3B) and trained with much more data. In comparison to unified models, Show-o also exhibits improvement. However, it is worth noting that FID on MSCOCO 30K may not be a comprehensively accurate assessment of generation fidelity. The reason lies in the fact that existing generation models are commonly fine-tuned with high-quality and aesthetic images that do not align with the distribution of the MSCOCO dataset.

**Results on GenEval.** One can observe in Table 3 that when comparing to the model in a similar size such as LDM (1.4B), Show-o obtains significantly better performance in all six metrics, with an improvement of around 0.24 overall. Besides, Show-o achieves a better performance than DALL·E 2, which is 5 times larger in model size, and SDXL. Further, Show-o, with only 1.3B parameters, achieves comparable performance to models with around two times larger number of parameters such as SD3 (2B). It indicates that our unified model's generative capabilities are comparable to or even surpass those of specialized generation models. In comparison to unified models such as CoDI, SEED-X, and Chameleon, Show-o also demonstrates significant improvements.

**Qualitative Results.** We show image samples generated by Show-o in Fig. 7. One can observe that Show-o is capable of generating diverse, interesting, and realistic visual content in a resolution of 512×512. For example, Show-o can generate a futuristic style of car, a highly detailed face, cute objects, and vivid scenery with vibrant contrast.

**Text-guided Inpainting and Extrapolation.** As mentioned, Show-o naturally supports text-based inpainting and extrapolation without requiring any fine-tuning. We illustrate examples in Fig. 8 (a). As shown on the top of the figure, given an input image and inpainting mask, Show-o can inpaint the original red trolley car to a blue sports car with sleek curves and tinted windows based on the user-provided text prompt. Specifically, we first tokenize the original image, mask those tokens to be inpainted, and then Show-o will gradually replace the masked tokens with predicted image tokens. Besides, Show-o is capable of extrapolating the original image horizontally/vertically based on the given text prompt. These cases significantly demonstrate the inherent advantages of Show-o over those autoregressive models for downstream applications.

## 4.4 MIXED-MODALITY GENERATION OF VIDEO KEYFRAMES AND CAPTIONS

Here, we explore the mixed-modality generation ability of Show-o based on the text descriptions and video keyframes in the GenHowTo dataset. Given a sequence of interleaved text descriptions and video keyframes (as shown at the bottom of Fig. 4), Show-o is trained to predict the next text tokens or keyframe tokens conditioning on all preceding tokens. Thus, Show-o can generate mixed-modality of text descriptions and video keyframes. Examining a single frame, these tokens are generated in a diffusion manner. When considering the modeling of long sequences, as subsequent keyframes are produced based on all preceding text and image information, this can also be viewed as a form of temporal auto-regressive modeling.

We have tried to train Show-o using instructional examples and present qualitative examples in Fig. 8 (b). For example, given a question "Can you guide me through making Avocado and Apple Juice", Show-o exhibits the capability to generate video keyframes with text descriptions related to the question. It is apparent that the generated keyframes are temporally consistent.

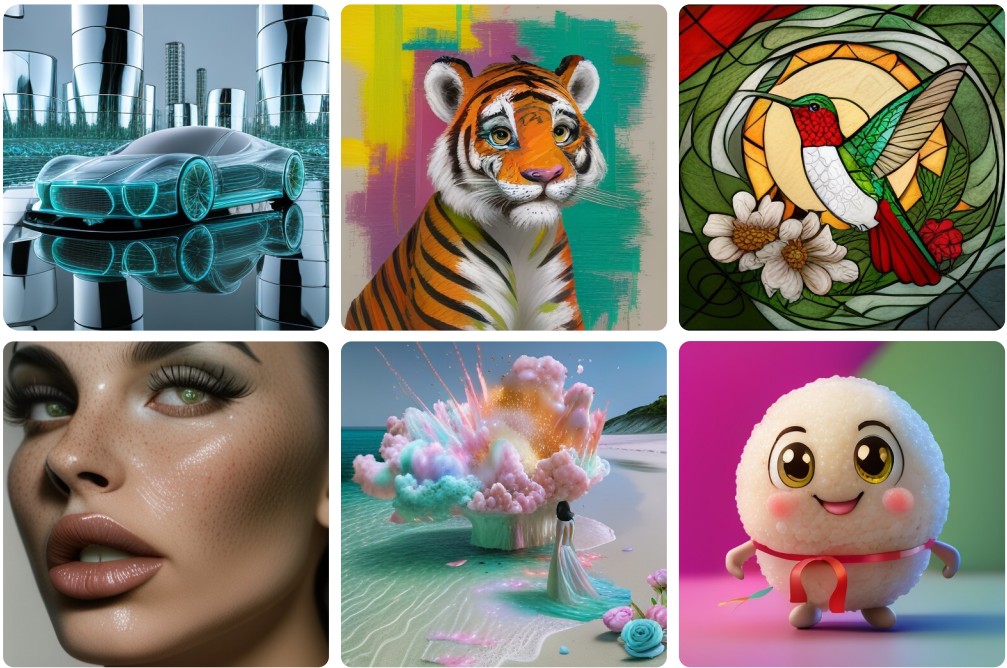

Figure 7: Images generated by Show-o. Text prompts are provided in Appendix G.

## 4.5 VIDEO UNDERSTANDING AND GENERATION

Beyond image understanding and generation, we have explored Show-o to support video understanding and generation. We inflate the existing pre-trained MAGVIT-v2 to support encoding videos into discrete tokens, compressing an 8 FPS video tensor of $3 \times 17 \times 256 \times 256$ into $5 \times 16 \times 16$. In this way, following the sequence format and omni-attention of multimodal understanding and generation introduced in Section 3.2, it is convenient to fine-tune the existing Show-o to involve video understanding and generation. One can observe visual examples in Figs. 8 (c) and (d) that Show-o can accurately comprehend and describe the variations in the video and generate consistent video frames with "a jeep car is approaching from a distance". More examples can be found in Appendix H.

## 4.6 ABLATION STUDIES

**Impact of dataset scale and image resolution on multimodal understanding.** As only discrete image tokens are extracted from the vision tokenizer, it is required to learn image token

Table 4: Impact of dataset scale and image resolution on the learning of discrete image token embeddings for multimodal understanding.

| # Image-text | Resolution | POPE | MME | Flickr30k | VQAv2$_{(test)}$ | GQA | MMMU |
|---|---|---|---|---|---|---|---|
| 35M | $256^2$ | 73.8 | 948.4 | 36.2 | 59.3 | 48.7 | 25.1 |
| 2.0B | $256^2$ | 76.2 | 1014.9 | 48.9 | 64.7 | 54.2 | 25.0 |
| 2.0B | $512^2$ | **80.0** | **1097.2** | **62.5** | **69.4** | **58.0** | **26.7** |

embeddings in Show-o from scratch. Unlike aligned image representations from the CLIP well-trained on a large-scale image-text dataset, Show-o necessitates the multimodal alignment between image and text embeddings during the pre-training stages. Here, we study the impact of the dataset scale and image resolution on the learning of discrete image token embeddings for multimodal understanding in Table 4. One can observe that the multimodal understanding capabilities of Show-o are consistently improved when increasing the data scale and image resolution. This reveals that it is required to involve more image-text pairs for multimodal alignment and more image tokens to represent an image for better comprehending the image information.

As illustrated in Fig. 3(a), the default Show-o adopts the pre-trained MAGVIT-v2 to tokenize input image to discrete tokens, which are then passed to the embedding layer to obtain embeddings as input for multimodal understanding. Beyond, we provide a systematic exploration of different design choices for the input of Show-o to enhance multimodal understanding. Specifically, as shown in Fig. 3(b) and (c), instead of discrete image tokens, we extract the continuous image representations

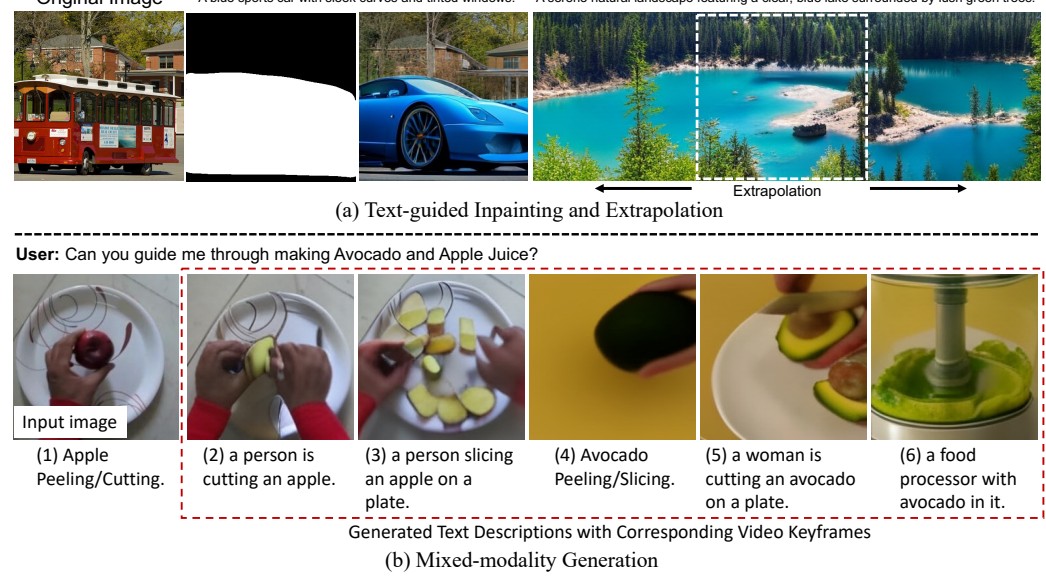

(a) Text-guided Inpainting and Extrapolation

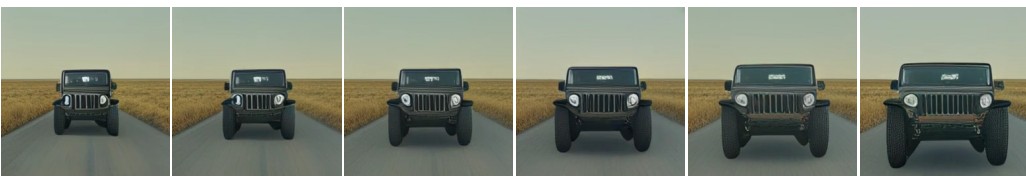

(b) Mixed-modality Generation

(c) Video Understanding

(d) Video Generation

Figure 8: Examples of text-guided inpainting/extrapolation, mixed-modality generation, video understanding and generation. 6 frames are sampled from the (generated) video for illustration.

from the pre-trained MAGVIT-v2 and CLIP-ViT, respectively, as input for Show-o when dealing with multimodal understanding. We provide experimental results and insights in Appendix I.

Additionally, we present qualitative examples to illustrate the impact of sampling steps and classifier-free guidance for text-to-image generation in Appendix I. In observation, increasing the sampling steps can allow the synthesis of an image that closely adheres to the prompt and improve fidelity. Besides, the classifier-free guidance can significantly make the colors and contents more diverse and consistent with the given text prompt.

We also discuss the failure modes of Show-o in Appendix K.

## 5 CONCLUSION

This paper proposed a unified transformer, *i.e.,* Show-o, to unify multimodal understanding and generation. Show-o for the first time unified autoregressive and (discrete) diffusion modeling that can handle different modalities in distinct ways. Extensive experimental results demonstrated that Show-o is comparable to even better than individual expert models across a wide range of vision-language tasks. This highlighted its potential as a next-generation foundation model.

## 6 ACKNOWLEDGMENTS

This research is supported by the National Research Foundation, Singapore under its AI Singapore Programme (AISG Award No:AISG3-RP-2022-030).

We would like to express our sincere gratitude to Henry Hengyuan Zhao for his valuable discussions and insightful feedback on multimodal understanding, Mingrui Wang for his patient assistance in helping us set up the development environment, and Han Yao for providing preprocessed video datasets.

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

Image corruption by adding different level mask (noise) tokens

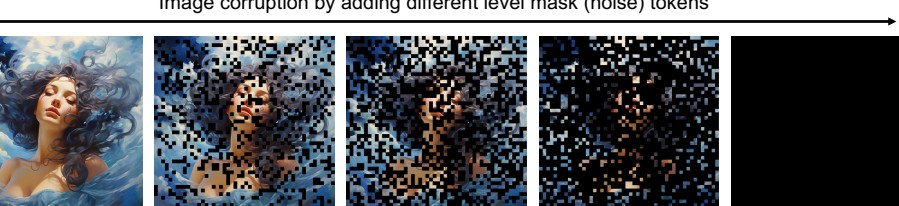

Image generation by iteratively removing mask (noise) tokens

Figure 9: Illustration of image corruption by adding different level mask (noise) tokens and image generation by iteratively removing mask (noise) tokens in the absorbing discrete diffusion paradigm.

## APPENDIX

## A PRELIMINARIES

In recent years, denoising diffusion probabilistic models (DDPMs) (Ho et al., 2020a) have demonstrated unprecedented performance in text-to-image/video generation in continuous state spaces, particularly exemplified by the popular Stable Diffusion series (Podell et al., 2023; Esser et al., 2024). Concurrently, discrete denoising diffusion probabilistic models (D3PMs) (Austin et al., 2021) have also shown impressive capabilities in modeling data in discrete form, featuring models like VQ-Diffusion (Gu et al., 2022) and Copilot4D (Zhang et al., 2024). Further, MaskGIT (Chang et al., 2022) and Muse (Chang et al., 2023) have demonstrated a simplified discrete diffusion that can effectively model discrete image tokens. Our Show-o model is built upon MaskGIT so that both such discrete visual and textual tokens can share a unified learning objective format. In the following, we provide preliminaries for diffusion models and draw the connection between discrete diffusion and mask token prediction employed in MaskGIT.

In diffusion models, the forward process $q(\mathbf{x}_{1:T}|\mathbf{x}_0) = \prod_{t=1}^{T} q(\mathbf{x}_t|\mathbf{x}_{t-1})$ corrupts the image data $\mathbf{x}_0 \sim q(\mathbf{x}_0)$ into latent variables $\mathbf{x}_1, \cdots, \mathbf{x}_T$ in different noise level. The reverse Markov process is learned to iteratively remove the noises added to the latent variables towards the real image distribution $q(\mathbf{x}_0)$. In the continuous scenario, the transition distribution $q(\mathbf{x}_t|\mathbf{x}_{t-1})$ is commonly characterized by a Gaussian distribution:

$$q(\mathbf{x}_t|\mathbf{x}_{t-1}) = \mathcal{N}(\mathbf{x}_t|\sqrt{1-\beta_t}\mathbf{x}_{t-1}, \beta_t\mathbf{I}), \qquad (4)$$

where the mean is $\sqrt{1-\beta_t}\mathbf{x}_{t-1}$ and the variance is $\beta_t$. For images tokenized into $K$ (*i.e.,* the codebook size) categorical random variables $\mathbf{x}_t, \mathbf{x}_{t-1} \in \{1, \cdots, K\}$ and given a $[\texttt{MASK}]$ state, the transition distribution is instead formulated by a stochastic transition matrix $\mathbf{Q}_t \in \mathbb{R}^{(K+1)\times(K+1)}$:

$$q(\mathbf{x}_t|\mathbf{x}_{t-1}) = \text{Cat}(\mathbf{x}_t|\mathbf{x}_{t-1}\mathbf{Q}_t), \qquad (5)$$

where $[\mathbf{Q}_t]_{ij} = q(\mathbf{x}_t = j|\mathbf{x}_{t-1} = i)$, $\mathbf{x}_{t-1}\mathbf{Q}_t$ indicates the row vector-matrix product, and $\text{Cat}(\mathbf{x}_t|\mathbf{x}_{t-1}\mathbf{Q}_t)$ is a categorical distribution over the one-hot row vector $\mathbf{x}_t$ given by $\mathbf{x}_{t-1}\mathbf{Q}_t$. When the transition matrix $\mathbf{Q}_t$ is applied to each image token in a sequence, the marginal and posterior at time step $t$ and $t-1$, respectively, are formulated as:

$$q(\mathbf{x}_t|\mathbf{x}_0) = \text{Cat}\left(\mathbf{x}_t|\mathbf{x}_0\overline{\mathbf{Q}}_t\right), \quad \text{where} \quad \overline{\mathbf{Q}}_t = \mathbf{Q}_1\mathbf{Q}_2\cdots\mathbf{Q}_t,$$

$$q(\mathbf{x}_{t-1}|\mathbf{x}_t, \mathbf{x}_0) = \frac{q(\mathbf{x}_t|\mathbf{x}_{t-1}, \mathbf{x}_0)q(\mathbf{x}_{t-1}|\mathbf{x}_0)}{q(\mathbf{x}_t|\mathbf{x}_0)} = \text{Cat}\left(\mathbf{x}_{t-1}\Big|\frac{\mathbf{x}_t\mathbf{Q}_t^\top \odot \mathbf{x}_0\overline{\mathbf{Q}}_{t-1}}{\mathbf{x}_0\overline{\mathbf{Q}}_t\mathbf{x}_t^\top}\right), \qquad (6)$$

where $q(\mathbf{x}_{t-1}|\mathbf{x}_t, \mathbf{x}_0) = q(\mathbf{x}_{t-1}|\mathbf{x}_t)$ because of the Markov property.

In the following, we introduce the **Absorbing-Uniform Discrete Diffusion** by defining the stochastic transition matrix $\mathbf{Q}_t$ as follows:

$$\mathbf{Q}_t = \mathbf{Q}_t^a\mathbf{Q}_t^u, \qquad (7)$$

where $\mathbf{e}_m$ is a one-hot vector with a value of 1 at the index of $[\texttt{MASK}]$ token, $\mathbf{Q}_t^a = (1-\alpha_t)\mathbf{I} + \alpha_t\mathbf{1}\mathbf{e}_m^\top$, and $\mathbf{Q}_t^u = \mathbf{I} - \beta_t(\mathbf{I} - \mathbf{e}_m\mathbf{e}_m^\top) + \frac{\beta_t}{(K+1)}(\mathbf{1} - \mathbf{e}_m)(\mathbf{1} - \mathbf{e}_m)^\top$. Here, $\alpha_t$ and $\beta_t$ represent the

probabilities of an image token transforming into the $[\texttt{MASK}]$ token and non-$[\texttt{MASK}]$ token at time step $t$, respectively. Specifically, the matrix form of $\mathbf{Q}_t$ can be written as:

$$
\mathbf{Q}_t = \begin{bmatrix} \omega_t + \nu_t & \nu_t & \nu_t & \cdots & \nu_t & \alpha_t \\ \nu_t & \omega_t + \nu_t & \nu_t & \cdots & \nu_t & \alpha_t \\ \nu_t & \nu_t & \omega_t + \nu_t & \cdots & \nu_t & \alpha_t \\ \vdots & \vdots & \vdots & \ddots & \vdots & \vdots \\ \nu_t & \nu_t & \nu_t & \cdots & \omega_t + \nu_t & \alpha_t \\ 0 & 0 & 0 & \cdots & 0 & 1 \end{bmatrix}, \tag{8}
$$

where $\omega_t = (1 - \alpha_t - \beta_t)$ and $\nu_t = \frac{\beta_t}{(K+1)}$. Intuitively, during the corruption process, each image token in the sequence has a probability of $\alpha_t$ to be replaced by the $[\texttt{MASK}]$ token, a chance of $\nu_t$ to be uniformly diffused, and a probability of $\omega_k + \nu_t$ remain unchanged. Besides, if a token turns into a $[\texttt{MASK}]$ token, it will stay in the same $[\texttt{MASK}]$ state during the following corruption process. Likewise, $\overline{\mathbf{Q}}_t = \overline{\mathbf{Q}}_t^a \overline{\mathbf{Q}}_t^u$ can be accordingly derived. An illustration of the image corruption process using $[\texttt{MASK}]$ token is provided in Fig. 9.

The evidence-lower bound (ELBO) for the variational diffusion models is:

$$
-\mathcal{L}_{\mathrm{ELBO}}(\mathbf{x}_0, \theta) = \mathbb{E}_{q(\mathbf{x}_{1:T}|\mathbf{x}_0)} \bigg[ -\underbrace{D_{\mathrm{KL}}[q(\mathbf{x}_T|\mathbf{x}_0) \,\|\, p(\mathbf{x}_T)]}_{\mathcal{L}_T} + \underbrace{\log p_\theta(\mathbf{x}_0|\mathbf{x}_1)}_{\mathcal{L}_0}
$$
$$
- \sum_{t=2}^{T} \underbrace{D_{\mathrm{KL}}[q(\mathbf{x}_{t-1}|\mathbf{x}_t, \mathbf{x}_0) \,\|\, p_\theta(\mathbf{x}_{t-1}|\mathbf{x}_t)]}_{\mathcal{L}_{t-1}} \bigg]. \tag{9}
$$

Considering the proposition in (Campbell et al., 2022) and the following parameterization of the reverse process:

$$
p_\theta(\mathbf{x}_{t-1}|\mathbf{x}_t) = \sum_{\mathbf{x}_0} q(\mathbf{x}_{t-1}|\mathbf{x}_t, \mathbf{x}_0) p_\theta(\mathbf{x}_0|\mathbf{x}_t), \tag{10}
$$

the variational lower bound can be further expressed under the image distribution $q(\mathbf{x}_0)$ (referring to the proof provided by Zhang et al. (2024) as detailed in the Appendix B):

$$
\mathbb{E}_{q(\mathbf{x}_0)}[\log p_\theta(\mathbf{x}_0)] \geq \mathbb{E}_{q(\mathbf{x}_0)}[-\mathcal{L}_{\mathrm{ELBO}}(\mathbf{x}_0, \theta)] \geq \sum_{t=1}^{T} \mathbb{E}_{q(\mathbf{x}_0)q(\mathbf{x}_t|\mathbf{x}_0)}[\log p_\theta(\mathbf{x}_0|\mathbf{x}_t)] + C. \tag{11}
$$

When deriving this lower bound, the discrete diffusion paradigm can be further simplified by restricting each image token to be either unchanged or replaced with the $[\texttt{MASK}]$ token, with no possibility of becoming other categorical variables. The resulting lower bound is effectively the Cross-Entropy loss used in MaskGIT (Chang et al., 2022), which is the mask token prediction to learn a neural network $p_\theta$ to reconstruct masked regions of $\mathbf{x}_0$ from the noised $\mathbf{x}_t$. In this work, we follow MaskGIT to integrate this simplified discrete diffusion paradigm into Show-o because of its simplicity. Further, Muse (Chang et al., 2023) has successfully scaled up such a paradigm for text-to-image models of 3B parameters using 460M image-text pairs.

## B    ALTERNATIVE LOWER BOUND FOR THE VARIATIONAL DIFFUSION

$$\mathbb{E}_{q(\mathbf{x}_0)}[\log p_\theta(\mathbf{x}_0)]$$

$$= \mathbb{E}_{q(\mathbf{x}_0)}[\log \int p_\theta(\mathbf{x}_0, \mathbf{x}_1 \cdots \mathbf{x}_T) \mathrm{d}\mathbf{x}_1 \cdots \mathbf{x}_T]$$

$$= \mathbb{E}_{q(\mathbf{x}_0)}\left\{ \log \mathbb{E}_{q(\mathbf{x}_{1:T}|\mathbf{x}_0)}\left[ \frac{p_\theta(\mathbf{x}_{0:T-1}|\mathbf{x}_T)}{q(\mathbf{x}_{1:T}|\mathbf{x}_0)}p(\mathbf{x}_T) \right] \right\}$$

$$\geq \mathbb{E}_{q(\mathbf{x}_0)q(\mathbf{x}_{1:T}|\mathbf{x}_0)}\left[ \log \frac{p_\theta(\mathbf{x}_{0:T-1}|\mathbf{x}_T)}{q(\mathbf{x}_{1:T}|\mathbf{x}_0)} + \log p(\mathbf{x}_T) \right]$$

$$= \mathbb{E}_{q(\mathbf{x}_{0:T})}\left[ \sum_{t\geq1}^{T} \log \frac{p_\theta(\mathbf{x}_{t-1}|\mathbf{x}_t)}{q(\mathbf{x}_t|\mathbf{x}_{t-1})} + \log p(\mathbf{x}_T) \right]$$

$$= \mathbb{E}_{q(\mathbf{x}_{0:T})}\left[ \sum_{t\geq1}^{T} \log p_\theta(\mathbf{x}_{t-1}|\mathbf{x}_t) + \log p(\mathbf{x}_T) - \sum_{t\geq1}^{T} \log q(\mathbf{x}_t|\mathbf{x}_{t-1}) \right]$$

$$= \mathbb{E}_{q(\mathbf{x}_{0:T})}\left[ \sum_{t\geq1}^{T} \log \sum_{\tilde{\mathbf{x}}_0} q(\mathbf{x}_{t-1}|\mathbf{x}_t, \tilde{\mathbf{x}}_0)\tilde{p}_\theta(\tilde{\mathbf{x}}_0|\mathbf{x}_t) \right] + \underbrace{\mathbb{E}_{q(\mathbf{x}_{0:T})}\left[ \log p(\mathbf{x}_T) - \sum_{t\geq1}^{T} \log q(\mathbf{x}_t|\mathbf{x}_{t-1}) \right]}_{C_1}$$

$$= \mathbb{E}_{q(\mathbf{x}_{0:T})}\left[ \sum_{t\geq1}^{T} \log \sum_{\tilde{\mathbf{x}}_0} \frac{q(\mathbf{x}_{t-1}, \tilde{\mathbf{x}}_0|\mathbf{x}_t)}{q(\tilde{\mathbf{x}}_0|\mathbf{x}_t)}\tilde{p}_\theta(\tilde{\mathbf{x}}_0|\mathbf{x}_t) \right] + C_1$$

$$= \mathbb{E}_{q(\mathbf{x}_{0:T})}\left[ \sum_{t\geq1}^{T} \log \sum_{\tilde{\mathbf{x}}_0} \frac{q(\tilde{\mathbf{x}}_0|\mathbf{x}_{t-1})}{q(\tilde{\mathbf{x}}_0|\mathbf{x}_t)} \overbrace{q(\mathbf{x}_{t-1}|\mathbf{x}_t)}^{q(\mathbf{x}_t|\mathbf{x}_{t-1})q(\mathbf{x}_{t-1})/q(\mathbf{x}_t)} \tilde{p}_\theta(\tilde{\mathbf{x}}_0|\mathbf{x}_t) \right] + C_1$$

$$\geq \mathbb{E}_{q(\mathbf{x}_{0:T})}\left[ \sum_{t\geq1}^{T} \sum_{\tilde{\mathbf{x}}_0} q(\tilde{\mathbf{x}}_0|\mathbf{x}_{t-1}) \log \left( \frac{q(\mathbf{x}_{t-1}|\mathbf{x}_t)}{q(\tilde{\mathbf{x}}_0|\mathbf{x}_t)}\tilde{p}_\theta(\tilde{\mathbf{x}}_0|\mathbf{x}_t) \right) \right] + C_1$$

$$= \mathbb{E}_{q(\mathbf{x}_{0:T})}\left[ \sum_{t\geq1}^{T} \sum_{\tilde{\mathbf{x}}_0} q(\tilde{\mathbf{x}}_0|\mathbf{x}_{t-1}) \log \tilde{p}_\theta(\tilde{\mathbf{x}}_0|\mathbf{x}_t) \right] + C_1 + \underbrace{\mathbb{E}_{q(\mathbf{x}_{0:T})}\left[ \sum_{t\geq1}^{T} \sum_{\tilde{\mathbf{x}}_0} q(\tilde{\mathbf{x}}_0|\mathbf{x}_{t-1}) \log \frac{q(\mathbf{x}_{t-1}|\mathbf{x}_t)}{q(\tilde{\mathbf{x}}_0|\mathbf{x}_t)} \right]}_{C_2}$$

$$= \sum_{t\geq1}^{T} \mathbb{E}_{q(\mathbf{x}_0, \mathbf{x}_{t-1}, \mathbf{x}_t)}\left[ \sum_{\tilde{\mathbf{x}}_0} q(\tilde{\mathbf{x}}_0|\mathbf{x}_{t-1}) \log \tilde{p}_\theta(\tilde{\mathbf{x}}_0|\mathbf{x}_t) \right] + C_1 + C_2$$

$$= \sum_{t\geq1}^{T} \mathbb{E}_{q(\mathbf{x}_0, \mathbf{x}_{t-1}, \mathbf{x}_t)q(\tilde{\mathbf{x}}_0|\mathbf{x}_{t-1})}[\log \tilde{p}_\theta(\tilde{\mathbf{x}}_0|\mathbf{x}_t)] + C_1 + C_2$$

$$= \sum_{t\geq1}^{T} \mathbb{E}_{q(\mathbf{x}_0|\mathbf{x}_{t-1})q(\mathbf{x}_t|\mathbf{x}_{t-1})q(\mathbf{x}_{t-1})q(\tilde{\mathbf{x}}_0|\mathbf{x}_{t-1})}[\log \tilde{p}_\theta(\tilde{\mathbf{x}}_0|\mathbf{x}_t)] + C_1 + C_2$$

$$= \sum_{t\geq1}^{T} \mathbb{E}_{q(\mathbf{x}_t|\mathbf{x}_{t-1})q(\mathbf{x}_{t-1}, \tilde{\mathbf{x}}_0)}[\log \tilde{p}_\theta(\tilde{\mathbf{x}}_0|\mathbf{x}_t)] + C_1 + C_2$$

$$= \sum_{t\geq1}^{T} \mathbb{E}_{q(\mathbf{x}_t, \mathbf{x}_0)}[\log \tilde{p}_\theta(\mathbf{x}_0|\mathbf{x}_t)] + C_1 + C_2$$

The constants $C_1$ and $C_2$ are:

$$C_1 = \mathbb{E}_{q(\mathbf{x}_{0:T})}\left[ -\sum_{t=1}^{T}\log q(\mathbf{x}_t|\mathbf{x}_{t-1}) + \underbrace{\log p(\mathbf{x}_T)}_{\text{Note that } p(\mathbf{x}_T)=q(\mathbf{x}_T)} \right]$$

$$= \mathbb{E}_{q(\mathbf{x}_{0:T})}\left[ -\sum_{t=1}^{T}\log q(\mathbf{x}_t,\mathbf{x}_{t-1}) + \sum_{t=0}^{T}\log q(\mathbf{x}_t) \right]$$

$$C_2 = \mathbb{E}_{q(\mathbf{x}_{0:T})}\left[ \sum_{t=1}^{T}\log q(\mathbf{x}_{t-1}|\mathbf{x}_t) \right] - \mathbb{E}_{q(\mathbf{x}_{0:T})}\left[ \sum_{t=1}^{T}\sum_{\tilde{\mathbf{x}}_0} q(\tilde{\mathbf{x}}_0|\mathbf{x}_{t-1})\log q(\tilde{\mathbf{x}}_0|\mathbf{x}_t) \right]$$

$$= \mathbb{E}_{q(\mathbf{x}_{0:T})}\left[ \sum_{t=1}^{T}\log q(\mathbf{x}_t,\mathbf{x}_{t-1}) - \sum_{t=1}^{T}\log q(\mathbf{x}_t) \right] - \sum_{t=1}^{T}\mathbb{E}_{q(\mathbf{x}_{0:T})q(\tilde{\mathbf{x}}_0|\mathbf{x}_{t-1})}[\log q(\tilde{\mathbf{x}}_0|\mathbf{x}_t)]$$

$$C_1 + C_2 = \mathbb{E}_{q(\mathbf{x}_{0:T})}[\log q(\mathbf{x}_0) - \textstyle\sum_{t=1}^{T}\log q(\mathbf{x}_0|\mathbf{x}_t)]$$

The alternative lower bound can be derived as:

$$\mathbb{E}_{q(\mathbf{x}_0)}[\log p_\theta(\mathbf{x}_0)] \geq \sum_{t=1}^{T}\mathbb{E}_{q(\mathbf{x}_t,\mathbf{x}_0)}[\log p_\theta(\mathbf{x}_0|\mathbf{x}_t)] + \mathbb{E}_{q(\mathbf{x}_{0:T})}\left[\log q(\mathbf{x}_0) - \sum_{t=1}^{T}\log q(\mathbf{x}_0|\mathbf{x}_t)\right]$$

$$= \sum_{t=1}^{T}\mathbb{E}_{q(\mathbf{x}_0)q(\mathbf{x}_t|\mathbf{x}_0)}[\log p_\theta(\mathbf{x}_0|\mathbf{x}_t)] + C.$$

This proof is provided by Zhang et al. (2024).

## C  TRAINING PIPELINE

Given that the embedding of image tokens is newly initialized, it necessitates large-scale pre-training to align for multimodal understanding and generation. Besides, Show-o eliminates the text encoder to extract text embeddings for text-to-image generation, which poses a significant challenge for achieving effective alignment between text and image content within one single transformer. To this end, we employ a three-stage approach to progressively and effectively train Show-o:

i) **Image Token Embedding and Pixel Dependency Learning**: We employ RefinedWeb (Penedo et al., 2023) dataset to train Show-o to maintain the language modeling ability. Meanwhile, ImageNet-1K dataset (Deng et al., 2009) and large-scale image-text pairs are adopted to train Show-o for class-conditional image generation and image captioning, respectively. Here, we directly leverage the class names from ImageNet-1K as textual inputs for learning class-conditional image generation. This stage primarily involves the learning of new learnable embeddings for discrete image tokens, pixel dependency for image generation, and alignment between image and text for image captioning.

ii) **Image-Text Alignment for Multimodal Understanding and Generation**: Building upon the pre-trained weights, we proceed to involve training of text-to-image generation on the image-text data instead of the ImageNet-1K. This stage mainly focuses on image and text alignment for both image captioning and text-to-image generation.

iii) **High-Quality Data Fine-tuning**: Lastly, we further refine the pre-trained Show-o model by incorporating filtered high-quality image-text pairs for text-to-image generation and instructional data for multimodal understanding and mixed-modality generation.

## D  INFERENCE DETAILS

In inference, two types of predictions, *i.e.,* text and image tokens, are involved in Show-o. In multimodal understanding, given the conditional image and questions, text tokens are auto-regressively sampled from the predicted tokens with higher confidence. In visual generation, given $N$ text tokens and $M$ [MASK] tokens as initial input, Show-o predict $M$ logits $\ell^t = \{\ell_i^t\}_{i=1}^{M}$ in parallel, where

$\ell_i^t \in \mathbb{R}^{1 \times (K+1)}$ and $t$ is the time step. Following the work (Chang et al., 2023), we compute both the conditional logit $\ell_c^t$ and unconditional logit $\ell_u^t$ for masked tokens. The final logit $\ell^t$ of each [MASK] token is obtained by the following equation with a guidance scale $w$:

$$\ell^t = (1 + w)\ell_c^t - w\ell_u^t. \tag{12}$$

For each [MASK] token $u_*$ at location $i$, we sample an image token $u_i^t$ from the codebook based on the predicted probability $p_i^t = \text{softmax}(\ell_i^t)$ and indicate its predicted score $s_i \in \mathbb{R}$ as the confidence of this sampled token. The confidence of the unmasked token is set as 1.0. Next, we compute the number of image tokens $m$ that should be re-masked based on the mask scheduling function $\gamma$, where $m = \lceil \gamma(\frac{t}{T})M \rceil$. More details of mask scheduling functions can be found in MaskGIT (Chang et al., 2022). Subsequently, we replace the predicted image tokens with [MASK] token $u_*$ based on the following metric:

$$u_i^{(t+1)} = \begin{cases} u_*, & \text{if } s_i < \text{sorted}_j(s_j)[m]. \\ u_i^t, & \text{otherwise.} \end{cases} \tag{13}$$

The resulting sequence, consisting of the remaining image tokens and [MASK] tokens, will be fed back to Show-o for the subsequent round of prediction until reaching the final time step $T$. The finalized image tokens are decoded by the image tokenizer into an image.

## E  DATASET DETAILS

Three types of data are adopted for training Show-o: i) **Text-only Data**: We employ the publicly available RefinedWeb dataset (Penedo et al., 2023) to preserve the text reasoning capabilities of the pre-trained LLM. This dataset comprises approximately 1 billion instances (equivalent to 968 million individual web pages) and totals 2.8 terabytes of curated text data. ii) **Image Data with Class Names**: Show-o learns pixel dependencies using 1.28M images sourced from the ImageNet-1K (Deng et al., 2009) dataset. iii) **Image-Text Data**: For pre-training tasks correspond to multi-modal understanding and generation, we assemble roughly 35M image-text pairs from the publicly available datasets including CC12M (Changpinyo et al., 2021), SA1B (Kirillov et al., 2023), and LAION-aesthetics-12M[†]. Additionally, we further increase the data scale to around 2.0B by incorporating DataComp (Gadre et al., 2024) and COYO700M (Byeon et al., 2022) with some filtering strategies. Note that, we employ ShareGPT4V (Chen et al., 2023) to re-caption these datasets. Additionally, around 1M internal image-text pairs serve as high-quality text-to-image generation datasets for the final fine-tuning. Following LLaVA-v1.5 (Liu et al., 2024b), we incorporate LLaVA-Pretrain-558K and LLaVA-v1.5-mix-665K for instruction tuning. Moreover, the GenHowTo dataset (Souček et al., 2024) is utilized for mixed-modality generation.

## F  IMPLEMENTATION DETAILS

We initially conduct joint training of Show-o using the RefinedWeb, a collection of image-text pairs, and the ImageNet-1K for language modeling, image captioning, and class-conditional image generation, respectively, over 500K steps. Subsequently, we replace the class-conditional generation with the training for text-to-image generation using the around 35M image-text pairs for an additional 1,000K steps. The base model is trained on 48 A100 (80GB) GPUs with a total batch size of 1,152. We employ the AdamW optimizer with a weight decay of 0.01, 5,000 steps of warm-up, and an initial learning rate of 1e-4 with a cosine scheduling. Finally, we fine-tune Show-o with around 1M internal high-quality image-text pairs and adhere to the configuration of LLaVA-v1.5 for instruction data tuning. Note that, the current version of Show-o is based on Phi-1.5 (Li et al., 2023). In the following experiment sections, the default Show-o employs discrete image tokens as input for both multimodal understanding and generation. Show-o[†] and Show-o[‡] indicate the use of continuous image representations from the pre-trained MAGVIT-v2 and CLIP-ViT (corresponding to options (b) and (c) in Fig. 3), respectively, for multimodal understanding and we discuss this exploration in Section 4.6.

Based on the pre-trained Show-o, we continue to train it on the 2.0B image-text pairs for 500K steps and then we increase the image resolution to $512 \times 512$ and train Show-o for an additional

---

[†] https://huggingface.co/datasets/dclure/laion-aesthetics-12m-umap

500K steps. Finally, we fine-tune Show-o with around 1M internal high-quality image-text pairs and adhere to the configuration of LLaVA-v1.5 for instruction data tuning.

## G  Text Prompts

"A 3D render of a futuristic car made of glass, driving through a city of mirrors."

"A colorful cartoon of a tiger camouflaged in an abstract art painting, its stripes merging with the wild brushstrokes. "

"The image features a stylized stained glass illustration of a hummingbird with vibrant colors, set against a backdrop of swirling patterns and a large sun. The composition includes floral elements and intricate details, creating a vivid and dynamic scene that emphasizes the beauty of the bird. The colors range from greens to reds, enhancing the lively and artistic aesthetic of the piece. "

"A 3D render of a surreal explosion scene on the shore of a beautiful white sand beach with crystal clear water. The explosion has a spatter of oil paint with pastel colors and a thick consistency. The explosion is in a quiet and serene environment. A beautiful Japanese woman with a dress compacted to the sea is seen. There are butterfly petals and flowers with an ethereal glow and bioluminescence. There are pink and blue roses, and the overall image has a surreal and dreamlike quality. "

"A hyper-realistic close-up photograph of a woman's face, focusing on the left side. The image is highly detailed and realistic, showing voluminous glossy lips slightly parted, a well-defined nose, and open eyes with long eyelashes that cast shadows on the skin. The eye color is crystal clear almond green. The skin texture is crisp, with incredible detail of natural, lush skin and pores and freckles, with subtle highlights and shadows that give a realistic, close-up appearance. "

"A 3D render of a cute, round rice ball character named Mochi, with big, sparkling eyes that convey curiosity and joy. Its body is a soft, fluffy white with a slight sheen, resembling freshly cooked rice. Mochi has small, rosy cheeks that give it a warm, friendly expression. A tiny smile brightens its face, and it often sports a colorful ribbon tied around its "waist," adding a playful touch. Mochi's arms and feet are cartoonishly short, allowing it to bounce adorably around its surroundings. This time, Mochi is placed against a background that is a vibrant explosion of colors, with bright hues of fuchsia, turquoise, lemon yellow, and emerald green creating a canvas of vibrant contrasts and playful energy. The clashing colors make Mochi's soft white body and rosy cheeks stand out even more, inviting viewers into a world of cheerful exuberance and visual delight."

## H  More Examples of Video

We provide more examples of video understanding and generation in Fig. 10.

## I  Ablation Studies

**Impact of Vision Encoder for Multimodal Understanding.**  The default Show-o employs MAGVIT-v2 to encode images into discrete tokens for both multimodal understanding and generation. Inspired by the literature (Liu et al., 2024b), we investigate the impact of the most popular design choice of vision encoder, *i.e.,* the pre-trained CLIP ViT (Radford et al., 2021), for multimodal understanding. We first compare the two settings using our Show-o model. In Table 5, the comparison between Exp 2 and Exp 4, Exp 3 and Exp 5 clearly demonstrates that continuous representations from CLIP-ViT have significantly better performance on multimodal understanding than that of MAGVIT-v2. This mainly attributes to: i) The CLIP-ViT is pre-trained on a much larger dataset (400M) than that of our pre-trained MAGVIT-v2 (35M); ii) In contrast to image reconstruction learning objective in MAGVIT-v2, the discriminative loss, *i.e.,* image-text matching, in CLIP-ViT makes the extracted representations easier to be adapted for multimodal understanding.

**Impact of Various Representations for Multimodal Understanding.** In typical multimodal understanding models like LLaVA, the image representation extraction and cross-modal alignment usually happen in the continuous space. However, image tokenizers such as MAGVIT-v2 naturally yield discrete image tokens. As shown in Table 5, we compare the two types of input, *i.e.,* continuous representations and discrete tokens, in the multimodal understanding scenario. In Exp 6

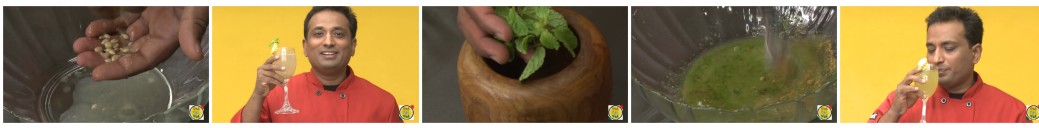

Describe the video content.

The video features a man in a red shirt who is making a drink by adding various ingredients to a glass. He pours the mixture into a strainer and then adds a lemon slice to the glass. The man then drinks the drink from the glass.

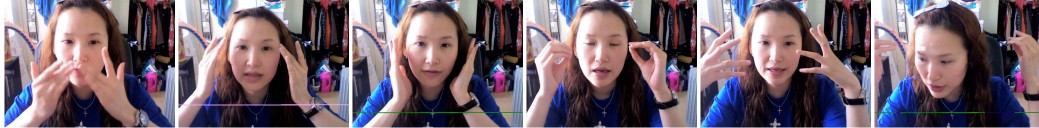

Describe the video content.

The video features a woman who is sitting in front of a computer and talking to the camera. She is wearing a blue shirt and has her hands in her lap. Throughout the video, she is seen making various gestures and speaking to the camera. The background of the video is filled with various items, including a clock, a mirror, and a shelf with clothes. The woman's expressions and gestures vary throughout the video, but they are all focused on her conversation.

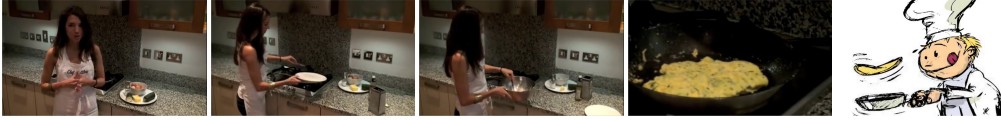

Describe the video content.

The video features a woman in a kitchen who is cooking an omelette. She is shown stirring the eggs in a bowl, adding butter and grated cheese, and then flipping the omelette. The video also includes a cartoon character who is shown cooking an omelette.

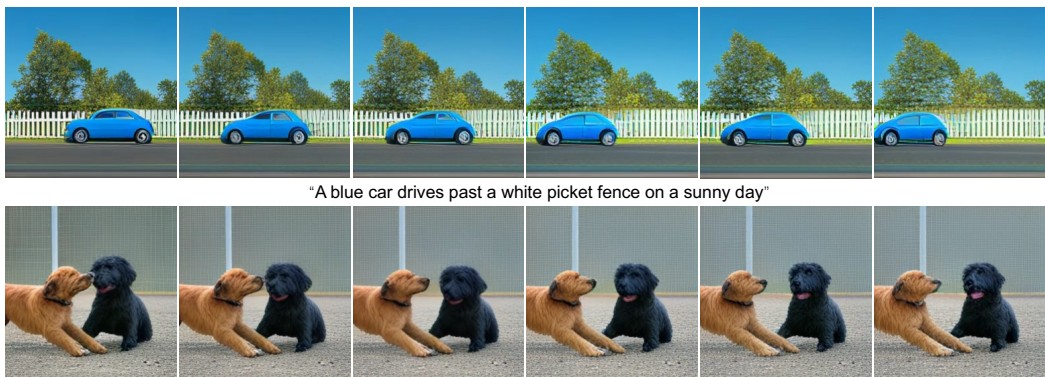

"A blue car drives past a white picket fence on a sunny day"

"There are two dogs, one brown and one black, playing together in a fenced-in area."

Figure 10: More examples on video understanding and generation. Only some key frames are sampled for illustration.

Table 5: Ablation studies of various vision encoders and kinds input representations for multimodal understanding. Note that this experiment is based on the Show-o pre-trained on 35M image-text data in a resolution of $256 \times 256$.

| # Exp | Method | Vision Encoder | Unified Pretrain | Feature type | POPE | MME | Flickr30k | VQAv2$_{(val)}$ | GQA | MMMU |
|-------|--------|----------------|------------------|--------------|------|-----|-----------|-----------------|-----|------|
| 1 | LLaVA | CLIP-ViT | ✗ | Continuous | 84.1 | 1128.0 | 69.6 | 73.0 | 56.5 | 30.67 |
| 2 | Show-o[‡] | CLIP-ViT | ✓ | Continuous | 84.5 | 1182.7 | 64.3 | 71.9 | 57.5 | 27.4 |
| 3 | Show-o[‡] | CLIP-ViT | ✗ | Continuous | 84.5 | 1161.6 | 68.5 | 73.5 | 58.7 | 29.2 |
| 4 | Show-o[†] | MAGVIT-v2 | ✓ | Continuous | 74.3 | 947.8 | 33.9 | 59.4 | 51.0 | 26.7 |
| 5 | Show-o[†] | MAGVIT-v2 | ✗ | Continuous | 65.1 | 800.0 | 12.3 | 50.8 | 43.9 | 24.6 |
| 6 | Show-o | MAGVIT-v2 | ✓ | Discrete | 73.8 | 948.4 | 36.2 | 57.8 | 48.7 | 25.1 |
| 7 | Show-o | MAGVIT-v2 | ✗ | Discrete | 63.8 | 689.1 | 4.5 | 46.1 | 40.5 | 28.1 |

and 7, we use the pre-trained MAGVIT-v2 to extract discrete tokens and train an embedding layer to embed the tokens into the continuous embedding space of the LLM. In Exp 4 and 5, we modify MAGVIT-v2 to output continuous representations without quantization. The cross-modal projection layer follows the setting of LLaVA. The comparison between Exp 5 and Exp 7 reveals that discrete

"A colorful cartoon of a tiger camouflaged in an abstract art painting, its stripes merging with the wild brushstrokes."

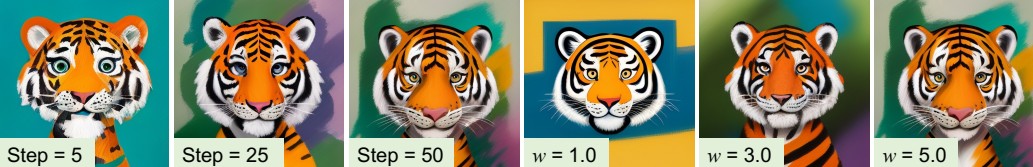

Figure 11: Illustration of generated samples using different sampling steps and classifier-free guidance scale $w$. This experiment is based on the Show-o in a resolution of $512 \times 512$.

(a) Failure Cases in Multimodal Understanding          (b) Failure Cases in Visual Generation

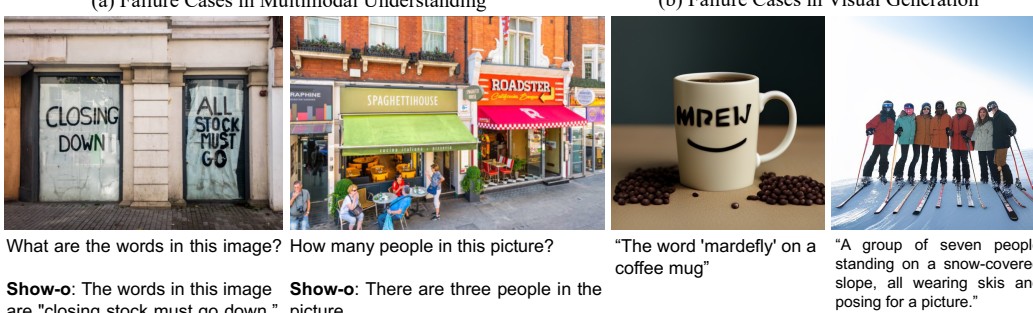

What are the words in this image? How many people in this picture?

**Show-o**: The words in this image are "closing stock must go down."

**Show-o**: There are three people in the picture.

"The word 'mardefly' on a coffee mug"

"A group of seven people standing on a snow-covered slope, all wearing skis and posing for a picture."

Figure 12: Illustration of failure cases of Show-o in multimodal understanding and generation.

tokens show much worse performance on most benchmarks. We attribute the performance gap to that popular multimodal understanding datasets, *e.g.,* LLaVA-Pretrain-558K, are not sufficient to align discrete image tokens into the language space, leading to an unsatisfactory cross-modal understanding. In contrast, continuous representations, already lying in a well-shaped embedding space, are much easier to align.

**Impact of Unified Pre-training for Multimodal Understanding.** Our training pipeline involves two-stage unified pre-training to learn image token embedding and image-text alignment for multimodal understanding and generation (as described in Section C). Here we elaborate on the impact of the unified per-training with different vision encoders and types of representations:

- CLIP-ViT with Continuous Representations. The comparison between Exp 2 and Exp 3 shows that the unified pre-training has a small negative effect on the CLIP ViT-based understanding, as the performance on most benchmarks has marginal degradations. We hypothesize that the MAGVIT-v2 token-based pre-training and the CLIP ViT-based tuning happen in nearly orthogonal dimensions, and the capability of the backbone has been spared to maintain the compatibility of the two tasks.

- MAGVIT-v2 with Continuous Representations. In the comparison between Exp 4 and Exp 5, we also notice a performance improvement brought by the unified pre-training, even though the pre-training uses discrete tokens while the experiments here use continuous features. This comparison further validates the hypothesis that unified pre-training enhances the multimodal understanding and reasoning capabilities of the backbone by diverse multimodal interactions during pre-training.

- MAGVIT-v2 with Discrete Tokens. The comparison between Exp 6 and Exp 7 shows that the unified pre-training has significantly boosted the multimodal understanding performance. This is intuitive since the pre-training also adopts MAGVIT-v2 discrete tokens as image representation. Specifically, we attribute the performance gain to that unified pre-training learns a better cross-modal alignment with large-scale data and enhances the multimodal understanding capabilities of the backbone.

**Impact of Sampling Steps.** We present generated results at $512 \times 512$ resolution with varying sampling steps on the left of Fig. 11. With just five steps, Show-o can produce an image that is roughly related to the given prompt. Increasing the sampling steps to 25 allows the synthesis of an image that closely adheres to the prompt. When the sampling step is set as 50, the generated image becomes more detailed and realistic. In contrast, auto-regressive models Team (2024); Sun

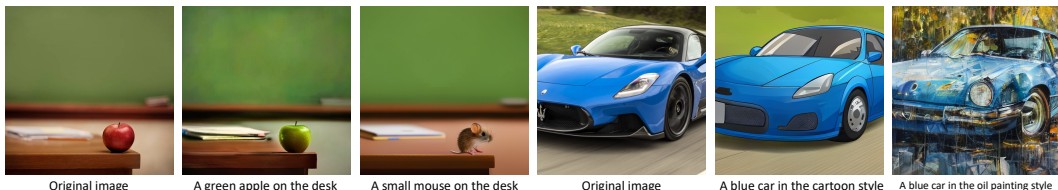

| Original image | A green apple on the desk | A small mouse on the desk | Original image | A blue car in the cartoon style | A blue car in the oil painting style |

Figure 13: Mask-free image editing.

et al. (2024) require 1024 sampling steps to generate an image of the same resolution when the downsampling rate is 16, which is around 20 times more steps than our approach.

**Impact of Classifier-free Guidance.** The visual variations of generated images with different classifier-free guidance scales $w$ are illustrated on the right of Fig. 11. It can be observed that the object in the generated images lacks detail without classifier-free guidance. As the classifier-free guidance scale $w$ is gradually increased to 3 and 5, the colors and contents become more diverse and consistent with the given text prompt.

## J  MASK-FREE IMAGE EDITING

Given an image, it can be converted into discrete image tokens, which can then be subject to iterative random masking for sampling purposes. This process allows for image editing without the need for predefined masks. For instance, as illustrated in Fig. 13, Show-o enables local-region modifications such as "changing the red apple to green" or "replacing the apple with a mouse." Furthermore, Show-o facilitates global style adjustments like "transforming the original image into a cartoon or oil painting style."

## K  FAILURE CASES

We provide failure cases of Show-o in multimodal understanding and generation in Fig. 12. The current version of Show-o sometimes cannot accurately recognize the text and count the object instances and exhibits challenges in generating correct belongings such as skis for each instance. One can observe that Show-o fails to identify the phrase "closing down" in the left of Fig. 12(a) and is unable to generate the term "mardefly" (as shown left of Fig. 12(b)). This limitation is mainly attributed to the insufficiency of specific data tailored to these scenarios, as our model relies on a limited set of image-text pairs sourced from publicly available datasets and utilizes automatically generated captions. Enriching such kind of data holds promise for addressing these failure modes in Show-o, an aspect that will be explored in the future.

