# OpenReview forum: "Show-o: One Single Transformer to Unify Multimodal Understanding and Generation"
_ICLR.cc/2025/Conference — ICLR 2025 Poster_

### Official Review · Reviewer_8ezu · 2024-10-23

**Soundness:** 3
**Presentation:** 3
**Contribution:** 3
**Rating:** 6
**Confidence:** 4

**Summary:**

The paper introduces Show-o, a unified transformer model designed for both multimodal understanding and generation. By integrating autoregressive and discrete diffusion modeling, Show-o handles diverse inputs and outputs across various modalities, enabling tasks such as visual question answering, text-to-image generation, text-guided inpainting/extrapolation, and mixed-modality generation. Show-o achieves comparable or superior performance to specialized models in benchmarks and supports various applications.

**Strengths:**

- The motivation is clear, the proposed method is both sensible and innovative, and the experimental results are extensive and promising.

- The writing is good and easy to follow.

**Weaknesses:**

1. The discrete diffusion modeling proposed in the paper is essentially similar to MaskGIT’s masked autoregression, rather than a new concept. It’s recommended to rephrase the description or clarify this concept at the beginning of the paper, as it could otherwise increase the readers’ learning burden.

2. Besides inpainting and extrapolation, have the authors tried using [Text tokens, Image tokens, Image tokens] to perform image-to-image tasks like image editing? Can the Show-o framework support such tasks?

3. When using continuous representations, Show-o may not be able to handle generation tasks. However, based on several multimodal understanding benchmarks, continuous representations seem to deliver better performance. From the perspective of unified performance metrics, Show-o currently appears less satisfactory in understanding tasks.

4. Could you provide some experimental analysis of Show-o's inference speed in generation tasks?

5. Does the current design of Show-o support multi-turn interaction? Can it continue text-based question answering and image generation in ongoing dialogues?

**Questions:**

See weaknesses.

---

### Official Review · Reviewer_3aSb · 2024-11-03

**Soundness:** 3
**Presentation:** 3
**Contribution:** 3
**Rating:** 6
**Confidence:** 4

**Summary:**

The paper introduces a model called Show-o, designed to unify multimodal understanding and generation tasks within one transformer. Unlike traditional models that focus on either autoregressive or diffusion-based generation, Show-o combines both approaches to handle a wide range of inputs across tasks, including visual question answering, text-to-image generation, and text-guided inpainting.

Show-o can process performs competitively on standard benchmarks, often matching or exceeding models that are specialized for either understanding or generation. Additionally, Show-o supports mixed-modality tasks, like creating video frames with text descriptions. Overall the model offers a unified approach to multimodal transformers, aiming to simplify handling different types of media in a single framework.

**Strengths:**

- The paper compares against a lot of existing baselines on a lot of benchmarks thus indicating dense evaluation.
- The results are strong and competitive on image-text understanding and generation tasks across strong baselines.
- Good presentation and writing, a lot of well-made figures makes the understanding of the paper easy.
- Extension to videos and mixed modality makes the paper more applicable

**Weaknesses:**

- The paper doesn't do a good job at convincing why should one use discrete diffusion on Images instead of AR like Chameleon. While the introduction motivates the use of diffusion in images by discussing the number of forward evals, there are no emperical experiments showcasing this benefit.

- The comparisions with baselines is difficult to asses due to the use of different training/evaluation sets for each of the baselines. A human eval, with human deciding the starting prompt might make the comparisions a lot more fair.

**Questions:**

- The paper discusses the use of continuous image tokenizer in a version of their model, however it's unclear to me as to how is the continuous tokenization used for image generation as Show-O does discrete diffusion? If it's only used for understanding tasks, is this model trained on less data?

- Can the authors perform human evals for understanding and generation tasks?

- For understanding and generation evals in Table 1 and 2, is classifier free guidance being used for baselines and Show-O. Further how does varying the guidance score affect the results.

---

### Official Review · Reviewer_vpwM · 2024-11-04

**Soundness:** 4
**Presentation:** 4
**Contribution:** 4
**Rating:** 8
**Confidence:** 5

**Summary:**

This paper introduces Show-o, a unified autoregressive model for multimodal understanding and generation. Show-o integrates autoregressive and discrete diffusion modeling with an omni-attention mechanism, allowing it to utilize and output multimodal content. This design enables Show-o to handle a wide range of vision-language tasks like visual question answering, text-to-image generation, inpainting, extrapolation, and video generation with better benchmark results.

**Strengths:**

- Solid experiments, comprehensive comparison with existing works.
- Investigate a new paradigm of combining auto-regressive and discrete diffusion modeling for unifying understanding and generation.
- Extensive ablations for scaling up data, resolution, and visual representation (discrete/continuous and different pertaining).
- The paper is well-written and easy to follow.

**Weaknesses:**

- The model incurs significant training costs, making it challenging to scale, with potentially slower inference speeds.
- The benefits of unifying understanding and generation in a single model are unclear, especially given its lower performance in both areas compared to specialized unimodal models.

**Questions:**

- Are there any synergies or conflicts that arise from unifying understanding and generation, e.g., does this approach improve instruction-following capacity for generation, or reasoning capacity for text-only and multimodal inputs? Also, is there any conflict between the objectives of generation and understanding?
- For the LLaVA-v1.5-Phi-1.5 baseline, please clarify if the same pretraining data were used for Show-o? (I noticed that the same SFT data appears to be used)
- Please elaborate on the motivation and potential applications for unifying these two modalities, given that the current performance on both tasks still lacks usability (compared to VLM and diffusion models).
- Additional benchmark results would be helpful. For example, FID on MS-COCO for image generation; MMBench, SEED for visual understanding (reported VQA benchmarks have strong variance or bias); GSM8K, MMLU, HumanEval for text-only tasks.

---

### Official Review · Reviewer_4YnC · 2024-11-04

**Soundness:** 3
**Presentation:** 3
**Contribution:** 3
**Rating:** 6
**Confidence:** 4

**Summary:**

The work introduces Show-o, a unified transformer model that can do multimodal understanding and generation in one shared transformer model. Show-o combines autoregressive and diffusion modeling to handle various input and output modalities effectively. It supports diverse vision-language tasks such as visual question-answering, text-to-image generation, and mixed-modality tasks, showcasing competitive performance across benchmarks. The paper highlights Show-o's potential as a foundational model for the next generation and mentions forthcoming release of associated code and models.

**Strengths:**

1. The writing is good and clear.

2. The paper introduces a compelling study on Show-o, a unified MLLM, which compares with various MLLM models, showcasing its promise in both multimodal understanding and generation tasks.

3. A wide array of experiments is conducted to showcase the effectiveness of Show-o.

**Weaknesses:**

1. This work lacks comparisons with other unified models like Vila-u, Transfusion, and Emu3...

2. Show-o appears to utilize special tokens like [mmu] for multimodal understanding and [T2I] for visual generation to handle different tasks separately. The design raises questions about its necessity, as it explicitly reveals task information. Should a new special token be introduced for each new task?

3. I'm curious about Show-o's ability to manage interleaved input and output.

**Questions:**

Please see the weakness

---

### Meta-Review · Area_Chair_jYkq · 2024-12-20

**Metareview:**

The paper introduces a unified transformer model Show-o that can do multimodal understanding and generation in one shared transformer model. The model combines autoregressive and discrete diffusion modeling to handle different input and output modalities. This approach with a 1.3B base model achieves decent results on various tasks such as visual question-answering and text-to-image generation.

This paper presents a new approach of combining autoregressive and diffusion modeling, which is an insightful and valuable exploration in the direction of unified multimodal learning.
The reviewers are all positive about this paper.
I recommend accepting this paper.

**Additional Comments On Reviewer Discussion:**

Reviewer 4YnC raises concerns about insufficient comparison, special token design, and ability to manage interleaved input and output
The score is maintained as 6 after rebuttal.

Reviewer vpwM raises concerns about high training cost, inference speed, issues about scaling up, conflicts between tasks, and more benchmarks.
The score is increased from 6 to 8 after rebuttal.

Reviewer 3aSb raises concerns about architecture choice, unfair comparison, the generation ability with continuous tokens, impact of  classifier-free guidance.
The score is maintained as 6 after rebuttal.

Reviewer 8ezu raises concerns about clarify over prior discrete diffusion modeling, continuous tokens for generation, inference speed.
The score is increased from 5 to 6 after rebuttal.

The ratings are mixed in the initial reviews. (6, 6, 6, 5), and becomes positive after the rebuttal. (6, 8, 6, 6)
The concerns are addressed by authors in rebuttal via more experiments, comparison and analysis.

---

### Decision · Program_Chairs · 2025-01-22

Accept (Poster)